# Characteristics and Utility of Fluorescein Breakup Patterns among Dry Eyes in Clinic-Based Settings

**DOI:** 10.3390/diagnostics10090711

**Published:** 2020-09-17

**Authors:** Chika Shigeyasu, Masakazu Yamada, Norihiko Yokoi, Motoko Kawashima, Kazuhisa Suwaki, Miki Uchino, Yoshimune Hiratsuka, Kazuo Tsubota

**Affiliations:** 1Department of Ophthalmology, Kyorin University School of Medicine, Tokyo 1818611, Japan; yamadamasakazu@ks.kyorin-u.ac.jp; 2Department of Ophthalmology, Kyoto Prefectural University of Medicine, Kyoto 6020841, Japan; nyokoi@koto.kpu-m.ac.jp; 3Department of Ophthalmology, Keio University School of Medicine, Tokyo 1608582, Japan; motoko-k@a3.keio.jp (M.K.); uchinomiki@yahoo.co.jp (M.U.); tsubota@z3.keio.jp (K.T.); 4Santen Pharmaceutical Co., Ltd., Osaka 5308552, Japan; kazuhisa.suwaki@santen.com; 5Department of Ophthalmology, Juntendo University Graduate School of Medicine, Tokyo 1138431, Japan; yoshi-h@tkf.att.ne.jp; 6Tsubota Laboratory, Inc., Tokyo 1600016, Japan

**Keywords:** clinical study, dry eye, fluorescein breakup pattern, tear film breakup time

## Abstract

(1) Background: To evaluate the characteristics of fluorescein breakup patterns (FBUPs) among patients with dry eye disease (DED) and efficacy of FBUPs as a diagnostic test for DED subgroups. (2) Methods: The study enrolled 449 patients with DED. FBUPs were categorized as follows: area break (AB), line break (LB), spot break (SB), dimple break (DB), and random break (RB). Schirmer value, fluorescein breakup time (FBUT), keratoconjunctival score, DED subgroups and subjective symptoms were examined. (3) Results: LB patients presented with short FBUT and high keratoconjunctival score. AB patients presented with short FBUT, high cornea and keratoconjunctival scores. SB patients were young with short FBUT. DB patients exhibited low keratoconjunctival score. RB patients were young, with long FBUT and low keratoconjunctival scores. Among DED subgroups, LB and AB constituted 74.7% of aqueous-deficiency dry eye (ADDE). SB and DB constituted 42.4% of short FBUT dry eye (short FBUT-DE). Post-test probabilities and positive likelihood ratios for ADDE were 58.7% and 1.63, respectively; those for short FBUT-DE were 46.3% and 2.02, respectively. Patients with SB and AB exhibited significantly severe subjective symptoms than other FBUPs. (4) Conclusions: FBUPs are associated with both objective findings and symptoms of DED and may be a clinical tool for identification of DED subgroups.

## 1. Introduction

Dry eye disease (DED) is defined as a multifactorial disease of the ocular surface characterized by a loss of homeostasis of the tear film [1]. The International Dry Eye Workshop refined this definition in 2017 based on findings from numerous studies [2,3]. Definitions of DED in Japan were also refined in 2016 [4], which have a consensus with the new DED definition by the Asia Dry Eye Society [5]. These definitions underscore the common perception of the importance of tear film instability.

Tear film instability is clinically presented as shortening of tear film breakup time (TBUT). Shortening of fluorescein breakup time (FBUT) has been reported in over 95% of DED patients in clinical research, such as in the Dry Eye Cross-Sectional Study in Japan (DECS-J) [6] and Osaka study [7], confirming that the common features of DED include shortening of TBUT and tear film instability. Tear film instability may be caused by any layer of the tear film, comprising the lipid layer [8], aqueous layer [9], and mucins of the ocular surface epithelium [10,11]. The assessment of dry eye subtype in addition to DE diagnosis is needed for selecting the appropriate treatment modality. The Asia Dry Eye Society has proposed three types of dry eye: aqueous-deficient, decreased wettability, and increased evaporation [12]. It is suggested that these three types coincide with the problems of aqueous, membrane-associated mucins, and lipid/secretory mucins, respectively. Although each component cannot be quantitatively evaluated with current technology, a practical diagnosis based on the patterns of fluorescein breakup has been proposed [12,13]. Fluorescein breakup patterns (FBUPs) can be classified into five different patterns, each of which reflects different pathophysiology [13,14,15].

Although non-invasive TBUT (NIBUT) has been recommended for the measurement of TBUT [16], FBUT is still widely used in clinical practice due to its simplicity. A recent study confirmed that FBUT has valid diagnostic accuracy [17]. In addition, when FBUT is tested, simultaneous judgment of FBUPs can be performed. Classification of FBUPs may reveal the primary factors responsible for tear film breakup, and the relationship between dry eye-related symptoms and ocular surface manifestations has been reported from specialized dry eye clinics in university hospitals [13]. However, the FBUPs among DED in general eye clinics and characteristics of FBUPs with regard to DED subgroups have yet to be elucidated. Therefore, this study was performed to evaluate the characteristics of FBUPs among patients with DED in clinics and to evaluate the diagnostic efficacy of FBUPs for identification of DED subgroups.

## 2. Materials and Methods

### 2.1. Study Subjects

We collected data from the DECS-J, which was a cross-sectional observational study conducted at 10 eye clinics in Japan [6,18,19]. All investigators at the study sites were specialists in ocular surface disorders and DED and belonged to the Japanese Dry Eye Society. To ensure the quality of the survey, two investigators’ meetings were held prior to the start of patient enrollment to discuss study protocols and examination procedures including judgment of FBUPs. We distributed an instructional digital versatile disc (DVD) that contained movie files with typical samples of FBUPs to each investigator.

Outpatients who were at least 20 years of age and were newly or previously diagnosed with DE were consecutively enrolled. The criteria for DED diagnosis defined by the Japan Dry Eye Society were used. The criteria were as follows: (1) at least one abnormal tear examination result (Schirmer I test ≤ 5 mm or FBUT ≤ 5 s), (2) abnormal results of ocular surface vital staining tests (fluorescein keratoconjunctival staining score ≥ 3), and (3) presence of DED symptoms [5]. Subjects who met two of the criteria (probable DE) and all three criteria (definite DE) were included in the study. Up to 50 patients were enrolled at each of the 10 study sites.

This study was conducted in accordance with the guidelines of the World Medical Association Declaration of Helsinki and the Ethical Guidelines for Medical and Health Research Involving Human Subjects in Japan. Subjects received a full explanation of the procedures and provided written informed consent before enrolment. The study protocol was approved by the Institutional Review Board (IRB) of Clinical Study, Ryogoku Eye Clinic, Tokyo, Japan. The study was registered in a public registration system (University Hospital Medical Information Network registry no. UMIN 000015890, approval date: 15 December 2014).

### 2.2. Clinical Assessment

#### 2.2.1. Tear Function Tests and Ocular Surface Evaluation

Ophthalmic examinations included assessment of FBUT, FBUPs, conjunctival and corneal vital staining with fluorescein sodium, measurement of FBUT, and the Schirmer I test.

Test strips containing fluorescein sodium (Fluores Ocular Examination Test Paper, Ayumi Pharmaceutical Co., Tokyo, Japan) were used for vital staining and FBUT measurement. Two drops of saline solution was used on the fluorescein strip and after shaking off the excessive saline, the tip of a wetted test strip was applied once to the inferior temporal tear meniscus, and patients were requested to blink several times to ensure adequate mixing of the fluorescein dye with tears. The time interval between the last complete blink and the appearance of the first corneal dark spot was measured with a stopwatch. Three natural blinks were performed between each measurement, and the mean of three measurements was recorded as FBUT, assessing FBUPs at the same time. The measurement was performed from the right eye to the left eye.

The FBUPs were categorized using Yokoi’s classifications [13]. Briefly, this classification categorizes the fluorescein breakup patterns into five types. Area break (AB) occurs when the aqueous tear volume is extremely diminished. Line break (LB) results from the simultaneous action between the drag of the aqueous tear by the spreading tear lipid layer and suction effects on the aqueous tear from the lower tear meniscus. Spot break (SB) and dimple break (DB) are considered to result from the impaired wettability of the corneal surface. Random break (RB) is considered to be related to increased evaporation. FBUPs were observed three times. The reproducibility of two or three FBUPs were determined as the diagnosis for FBUPs. If all three observations varied, they were determined as undefined.

Corneal epithelial damage was then evaluated via corneal fluorescein staining according to the National Eye Institute grading system [2]. Briefly, corneal staining was graded with a score of 0 to 3 assigned to each of five corneal zones (superior, nasal, central, inferior, and temporal), with a maximum total score of 15. Corneal and conjunctival epithelial damage was determined according to the modified grading system of van Bijsterveld [20], wherein each eye was divided into three sections (temporal conjunctiva, cornea, and nasal conjunctiva) and scored from 0 to 3. The final score ranged from 0 (minimum) to 9 (maximum).

The Schirmer I test was performed without topical anesthesia after all other examinations had been completed. The Schirmer I test was selected as it is regarded as a standardized test that provides an estimation of stimulated reflex tear flow, according to the Dry Eye WorkShop (DEWS) reports [16]. A Schirmer test strip (Ayumi Pharmaceutical Co., Tokyo, Japan) was placed for 5 min at the outer third of the temporal lower conjunctival fornix. The strip was then removed, and the length of filter paper that had been wetted was recorded in millimeters. To avoid any effect on the Schirmer I test by keratoconjunctival staining, the tests were performed at a minimum of 15 min apart.

#### 2.2.2. Classifications of DED Subgroups

The classification of major DED subgroups was defined by the study site investigators based on patients’ background, clinical findings, and subjective symptoms [6]. The following criteria were proposed to the study site investigators. For the aqueous-deficient dry eye (ADDE) subgroup (Sjögren syndrome and non-Sjögren type aqueous tear deficient DED), Schirmer I test ≤ 5 mm and subjective symptoms had to be fulfilled. Those for the short fluorescein breakup time dry eye (FBUT-DE) subgroup were FBUT ≤ 5 s, Schirmer I test > 5 mm, and kerato-conjunctival score < 3. The Meibomian gland dysfunction (MGD) subgroup was diagnosed according to the Japanese criteria (abnormal findings around the orifice of the meibomian gland such as vascular engorgement, anterior or posterior replacement of the mucocutaneous junction, and irregular lid margin; orifice obstruction including plugging, pouting, and ridge; secretion by compression classified as hyper-, normal, and hypo-secretion) [21], which was performed on one-third of the central site of the upper lid. Friction-related conditions were determined as presence or absence of lid wiper epitheliopathy (upper and lower lid), conjunctivochalasis (on the lower temporal, central, and nasal sites), and superior limbic keratoconjunctivitis (SLK). Minor DED subgroups were visual display terminal (VDT)-related, drug-induced, incomplete blinking, oral medicine-induced, nerve palsy, and others.

#### 2.2.3. Subjective Eye Symptoms Questionnaire

We used the Dry Eye-Related Quality-of-Life Score (DEQS) developed and validated in Japan to assess dry eye symptoms [22]. The DEQS consists of 15 questions and is scored and summarized by a single summary score. The summary score derived from DEQS was considered to be a quantitative measure of DED symptoms, whereby 0 indicated the best and 100 indicated the worst. Positive symptoms of DED were determined when subjects responded affirmatively to one or more of the six questions on the Bothersome Ocular Symptoms subscale.

### 2.3. Statistical Analyses

We used a generalized linear mixed model, taking into consideration the right and left eyes for analysis. Data for parameters are presented as estimate ± standard error (SE) unless otherwise indicated. All statistical analyses were performed using SAS software, version 9.4 (SAS Inc., Cary, NC, USA). For comparisons between groups, Fisher’s exact test was used for dichotomous variables and Student’s *t*-test for continuous variables. Likelihood ratio was used to calculate posttest probabilities for ADDE and short FBUT-DE subgroups. *p*-values less than 0.05 were considered statistically significant for all analyses.

## 3. Results

### 3.1. Study Population

Forty-five to 50 patients were enrolled from each site and a total of 926 eyes of 463 patients were initially registered (Table 1). Of these, 898 eyes of 449 patients (63 men and 386 women; mean ± standard deviation [SD] age, 62.6 ± 15.7 years) met the inclusion criteria and were enrolled. Reasons for exclusion were disqualification in nine patients, younger than 20 years in three patients, and withdrawal of consent in two patients. In addition, records of FBUPs were not recorded in 31 eyes of 21 patients (20 eyes of 10 patients and 11 eyes of 11 patients). Accordingly, 867 eyes of 439 patients (60 men and 379 women; mean ± SD age, 62.5 ± 15.8 years) were enrolled in the final analysis.

### 3.2. FBUPs in Relation to Tear Function Tests and Ocular Surface Evaluation

Classifications of FBUPs were made in 867 eyes (Figure 1). Then 448 eyes (51.6%) were classified into LB, followed by 141 eyes (16.3%) into DB, 117 eyes (13.5%) into RB, 91 eyes (10.5%) into SB, 40 eyes (4.6%) into AB, and 30 eyes (3.5%) into undefined. Comparison of FBUP groups (LB, AB, SB, DB, and RB) indicated that subjects with SB and RB were significantly younger than those with LB.

Comparing FBUT among FBUP groups (LB: 2.9 ± 0.1, AB: 2.5 ± 0.2, SB: 2.0 ± 0.1, DB: 3.1 ± 0.1, and RB: 3.6 ± 0.1), statistically significant differences were observed in patients with LB and SB (*p* < 0.0001), LB and RB (*p* < 0.0001), AB and DB (*p* = 0.015), AB and RB (*p* < 0.0001), SB and DB (*p* < 0.0001), SB and RB (*p* < 0.0001), and DB and RB (*p* = 0.003), respectively (Figure 2). FBUT was shortest in SB and AB followed by LB and DB, and longest in RB (Figure 2A). No statistically significant differences were detected in Schirmer’s test I (mm) among FBUP groups (LB: 9.4 ± 0.5, AB: 9.8 ± 1.4, SB: 9.4 ± 0.9, DB: 11.1 ± 0.9, and RB: 10.5 ± 0.9) (Figure 2B).

With regard to corneal staining scores (0–15) among FBUP groups (LB: 3.4 ± 0.2, AB: 5.5 ± 0.4, SB: 3.5 ± 0.3, DB: 3.0 ± 0.3, and RB: 2.8 ± 0.3), statistically significant differences were observed in patients with LB and AB (*p* < 0.0001), LB and RB (*p* = 0.032), AB and SB (*p* < 0.0001), AB and DB (*p* < 0.0001), and AB and RB (*p* < 0.0001). Cornea staining score was the highest in AB, followed by LB. No significant differences between SB, DB, and RB were observed (Figure 2C). Comparison of keratoconjunctival staining scores (0–9) among FBUP groups (LB: 2.6 ± 0.1, AB: 4.0 ± 0.2, SB: 2.6 ± 0.2, DB: 2.3 ± 0.2, and RB: 2.2 ± 0.2) revealed statistically significant differences in patients with LB and AB (*p* < 0.0001), LB and DB (*p* = 0.044), LB and RB (*p* = 0.010), AB and SB (*p* < 0.0001), AB and RB (*p* < 0.0001), AB and DB (*p* < 0.0001), and SB and RB (*p* = 0.019), respectively. Keratoconjunctival staining score was the highest in AB, followed by LB and SB. No significant differences between DB and RB were observed (Figure 2D).

### 3.3. FBUPs in Relation to DED Subgroups

Classification of FBUPs according to the major DED subgroups indicated 66 eyes with Sjögren syndrome, 305 eyes with non-Sjögren type aqueous deficiency, 238 eyes with short FBUT, 66 eyes with MGD, 67 CL-related eyes, and 54 friction-related eyes. The 71 eyes classified in the other minor DED subgroups were excluded from further analysis. Sjögren syndrome and aqueous deficiency, which are representative types of ADDE, constituted 75.0% of AB and 57.2% of LB (Table 2). In contrast, short FBUT-DE, which is accompanied by impaired wettability of the corneal surface, constituted 38.3% of SB and 51.9% of DB. RB (83 eyes) comprised five eyes with Sjögren syndrome (6.0%), 21 eyes with aqueous deficiency (25.3%), 15 eyes with short FBUT (18.1%), 14 eyes with MGD (16.9%), 15 CL-related eyes (18.1%), and 13 friction-related eyes (15.7%). Overall, MGD, CL-related, and friction-related DED, which is accompanied by increased evaporation, constituted 50.7% of RB.

### 3.4. FBUPs of ADDE and Short FBUT-DE Subgroups

The two representative DED subgroups in our study were ADDE and short FBUT-DE. Among ADDE (371 eyes), 247 eyes (66.6%) were classified into LB, 32 eyes (8.6%) into SB, 30 eyes (8.1%) into AB, 91 eyes (10.5%) into AB, and 26 eyes (7.0%) into RB and DB each, respectively (Table 3). Overall, LB and AB constituted 74.7% of ADDE. Among the short FBUT-DE (238 eyes), 116 eyes (48.7%) were classified into LB, 70 eyes (29.4%) into DB, 31 eyes (13.0%) into SB, 15 eyes (6.3%) into RB, and two eyes (0.8%) into AB, respectively. Overall, SB and DB constituted 42.4% of short FBUT-DE. Pre-test probability, post-test probability, and positive likelihood ratio for ADDE (LB and AB) were 46.6%, 58.7%, and 1.63, respectively; those for short FBUT-DE (SB and DB) were 29.9%, 46.3%, and 2.02, respectively (Table 4).

### 3.5. FBUPs in Relation to Subjective Eye Symptoms Questionnaire

Comparison of DEQS score among each FBUP group (mean ± SD, LB: 26.7 ± 21.1, AB: 30.7 ± 23.5, SB: 33.8 ± 22.4, DB: 22.7 ± 17.1, and RB: 26.9 ± 18.2) revealed statistically significant differences in patients with LB and SB (*p* = 0.002), LB and DB (*p* = 0.048), AB and DB (*p* = 0.029), SB and DB (*p* < 0.0001), and SB and RB (*p* = 0.013). DEQS score was severe in SB and AB, followed by LB. No significant differences were detected between DB and RB (Figure 3).

## 4. Discussion

Stable preocular tear film is essential to maintain ocular health, for the visual system, and to protect and lubricate the ocular surface [23,24]. Normal tear film undergoes “breakup” within 10 or 20 s, unless it is re-established by blinking. However, disorder of the tear film and ocular surface, represented by DED, may cause rapid “breakup” of the tear film, which is described as instability of the tear film. This is observed as the rapid appearance of regions of localized drying. Of several techniques used to visualize “breakup” [25,26,27], the use of fluorescein is convenient and of great practical value to the ophthalmologist in general clinics. Measurements of FBUT result in highly reliable DED diagnoses. Simultaneous judgment of FBUPs may reveal the primary factors responsible for tear film breakup and may assist the diagnosis of DED subgroups [13]. However, classification of FBUPs has only been reported in specialized clinics for dry eyes in university hospitals. Thus, this study evaluated the clinical utility of FBUPs in general clinics.

Here, we report the distribution of FBUPs among dry eye patients in general clinics for the first time. The distribution in our study was 51.6% LB, 16.3% DB, 13.5% RB, 10.5% SB, 4.6% AB, and 3.5% undefined. Approximately half of the patients were classified as LB. Of the enrolled patients in our study, the average FBUT was 3.0 ± 1.6 s (mean ± SD), and 94.9% were FBUT ≤ 5 s. These results confirmed that a common characteristic of DED is the shortening of TBUT, caused by the instability of tear film. FBUPs were obtained and classified in majority of cases (439 patients; 97.8%). These results suggest that the classification of FBUPs was possible in general clinics.

The Asia Dry Eye Society has proposed three major tear pathophysiologies of tear breakup for tear film instability: aqueous deficiency, decrease in wettability of the corneal surface, and increased evaporation [12]. In addition, Yokoi and associates reported an association between the primary factors responsible for the five FBUPs and DED subgroups; AB and LB consisted mainly of Sjögren syndrome and non-Sjögren type aqueous deficiency, which accompanies aqueous tear deficiency. Spot break and dimple break consisted mainly of short FBUT-DE, which was accompanied by a decrease in wettability of the corneal surface which is supported by membrane-associated mucins, especially longest MUC16 [28]. Random breaks consisted mainly of MGD and CL-related, which were accompanied by an increase in evaporation [15]. We observed an association of FBUPs between the pathophysiology of tear breakup in DED subgroups: aqueous tear deficiency, decrease in wettability of the corneal surface, and increased evaporation. With regard to the characteristics, tear function, and ocular surface features of FBUPs, statistically significant differences were observed in each FBUP. Patients with AB were characterized by shorter FBUT and higher corneal and keratoconjunctival scores. Patients with SB presented with shorter FBUT. Results of objective tests in our study appear to reflect the theoretical formation of FBUPs and support the mechanism of each FBUP, as proposed by Yokoi et al. [13,14,15].

When each FBUP was classified into DED subgroups, the pathophysiologic characteristics of DED subgroups appeared to reflect the constitution of the FBUPs. Sjögren syndrome and non-Sjögren type aqueous deficiency, which are accompanied by aqueous deficiency, constituted 75.0% of AB and 57.2% of LB. Short FBUT-DE, which is accompanied by decreased wettability of the corneal surface, constituted 38.3% of SB and 51.9% of DB. MGD, CL-related, and friction-related DED, which are accompanied by increased evaporation, constituted 50.7% of RB.

Further analysis was performed in the two representative DED subgroups (ADDE and short FBUT-DE) to assess whether FBUPs would be beneficial for assisting the diagnosis of DED subtypes. Among the ADDE, LB and AB, which are characterized by aqueous tear deficiency, constituted 74.7%. The short FBUT-DE, SB, and DB, which are characterized by decreased wettability of the corneal surface, constituted 42.4%. The positive likelihood ratio of FBUPs for ADDE (LB and AB) and short FBUT-DE (SB and DB) were 1.63 and 2.02, respectively. This indicates that when FBUP of the examined eye is diagnosed as SB or DB, the probability of short FBUT-DE change from 29.9% to 46.3%.

Pretest probabilities, posttest probabilities, and positive likelihood ratio were 46.6%, 58.7%, and 1.63, respectively, for ADDE (LB and AB); and 29.9%, 46.3%, and 2.02, respectively, for short FBUT-DE (SB and DB) (Table 4). Although the positive likelihood ratios of FBUPs in our study were inadequate as a diagnostic test, this method is simple and can be performed in clinics. Thus, FBUPs appear to be beneficial as auxiliary measures for the classification of DED subtypes.

Among FBUP groups, DEQS was most severe in SB and AB, followed by LB. No significant differences between DB and RB were observed. Based on our results, SB mainly consisted of short FBUT-DE (38.3%), followed by ADDE (30.9%). AB mainly consisted of ADDE (60.0%), followed by Sjögren’s syndrome (15.0%). As previously reported, the subjective symptoms of short FBUT-DE are equally severe as those of ADDE [19,29,30].

Our study had several limitations. The rate of women was high and covered 86.4% in this study. Although the prevalence of dry eye disease is known to be higher in women compared to men, further analysis was performed to see the gender differences in FBUPs. The results were similar among the groups, and the order of the FBUPs were the same; line break (man: 54.2%, woman: 51.5%), dimple break (man: 12.7%, woman: 16.9%), random break (man: 11.9%, woman: 13.8%), spot break (man: 6.8%, woman: 11.1%), area break (man: 5.9%, woman: 4.4%), and undefined (man: 8.5%, woman: 2.7%), respectively. It resulted that the order of frequently diagnosed FBUPs seems to match between man and woman. Secondly, although automated NIBUT is recommended to measure TBUT rather than subjective FBUT by the observer [16], we were not able to measure them since not all the clinics had the device. Much attention was taken not to influence the tear film stability during the FBUT measurements; however, we must take in mind that NIBUT is known to be less affective. Thirdly, although all investigators in our study were trained by meetings and movie files to ensure consistent classification of FBUPs, there may have been variation in their evaluations. In addition, classification of FBUPs is progressing. In particular, “rapid expansion” of the breakup reflecting a decrease in wettability of the corneal surface is now being considered [14]. Further, there may have been modifications to the clinical presentation, as patients who had already been treated for DE were included in the study.

In summary, we have demonstrated the association of FBUPs with ocular surface features and tear function. Classification of FBUPs is a simple and convenient method that can be performed in clinics and may assist the identification of DED subgroups and selection of appropriate treatment modalities.

## Figures and Tables

**Figure 1 diagnostics-10-00711-f001:**
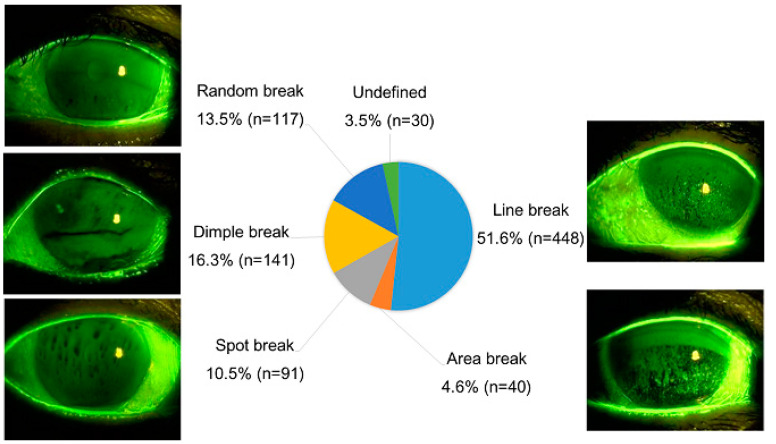
Distribution of fluorescein breakup patterns among dry eye patients. Among 867 eyes evaluated, approximately half (51.6%) were classified into line break.

**Figure 2 diagnostics-10-00711-f002:**
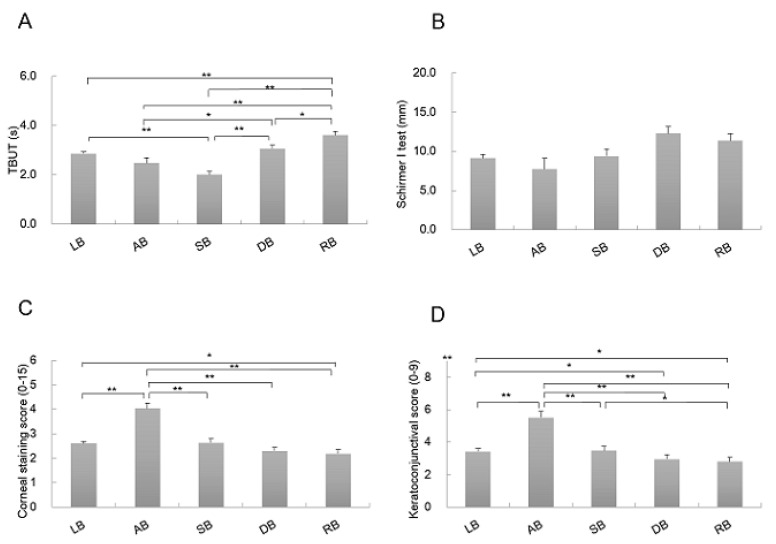
Results of fluorescein breakup patterns of tear examination and ocular surface results. Each graph shows the results of FBUPs of (**A**) FBUT, (**B**) Schirmer I test, (**C**) cornea staining score (National Eye Institute grading system scoring system), and (**D**) keratoconjunctival staining score (van Bijsterveld staining system). AB = area break; DB = dimple break; FBUPs = fluorescein breakup patterns; LB = line break; RB = random break; SB = spot break; FBUT = fluorescein breakup time. Data are estimate ± standard error. * *p* < 0.05, ** *p* < 0.01 between two comparisons.

**Figure 3 diagnostics-10-00711-f003:**
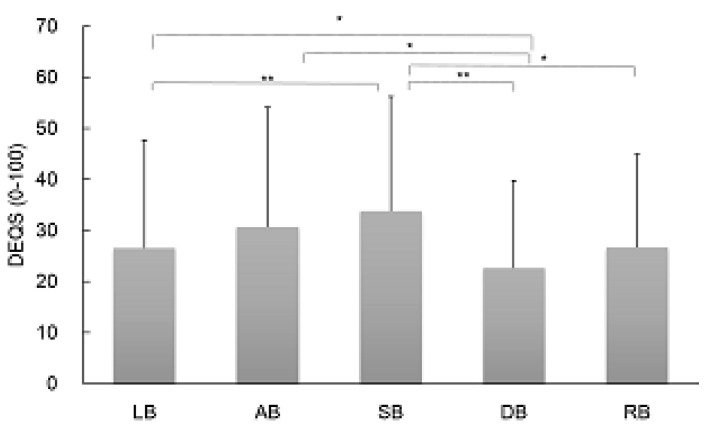
Results of fluorescein breakup patterns of Dry Eye Related Quality of Life Score. AB = area break; DB = dimple break; DEQS = Dry Eye Related Quality of Life Score; FBUPs = fluorescein breakup patterns; LB = line break; RB = random break; SB = spot break. Data are mean ± standard deviation. * *p* < 0.05, ** *p* < 0.01 between two comparisons.

**Table 1 diagnostics-10-00711-t001:** Characteristics of FBUPs of the study population. Among 867 eyes included in the final analysis, approximately half (51.6%) were classified as line break.

	Patients Eyes (%)	Age (yrs)	*p* Value (vs. Line)	Woman Eyes (%)	*p* Value (vs. Line)
All	867 (100)	62.5 ± 15.8		749 (86.4)	
Line break	448 (51.6)	64.3 ± 14.6	N/A	384 (85.7)	N/A
Area break	40 (4.6)	65.3 ± 17.6	0.71	33 (82.5)	0.58
Spot break	91 (10.5)	57.4 ± 16.1 *	0.0001	83 (91.2)	0.16
Dimple break	141 (16.3)	61.6 ± 17.5	0.07	126 (89.4)	0.27
Random break	117 (13.5)	59.0 ± 16.0 *	<0.01	103 (88.0)	0.52
Undefined	30 (3.5)	64.4 ± 15.6	0.99	20 (66.7) *	<0.01

FBUPs = fluorescein breakup patterns, N/A = not applicable. Data are mean ± standard deviation. * *p* < 0.05 compared with line break. *p* values are adjusted for age and gender.

**Table 2 diagnostics-10-00711-t002:** Tear breakup patterns and dry eye subgroups.

	Sjögren Syndrome	Aqueous Deficiency	Short FBUT	MGD	CL	Friction
Line break	10.2	47.0	26.9	5.6	6.3	4.2
Area break	15.0	60.0	5.0	2.5	12.5	5.0
Spot break	8.6	30.9	38.3	6.2	7.4	8.6
Dimple break	1.5	17.8	51.9	13.3	8.9	6.7
Random break	6.0	25.3	18.1	16.9	18.1	15.7
Undefined	8.0	32.0	16.0	16.0	8.0	20.0

DED = dry eye disease; CL = contact lens; MGD = meibomian gland dysfunction; short FBUT-DE = short fluorescein breakup time dry eye. The results are shown in percentage (%).

**Table 3 diagnostics-10-00711-t003:** Tear breakup patterns of aqueous deficient dry eye and short fluorescein breakup time dry eye subgroups.

	Area Break	Line Break	Spot Break	Dimple Break	Random Break	Undefined
ADDE	8.1	66.6	8.6	7.0	7.0	2.7
short FBUT-DE	0.8	48.7	13.0	29.4	6.3	1.7

ADDE = aqueous deficient dry eye; short FBUT-DE = short fluorescein breakup time dry eye. The results are shown in percentage (%).

**Table 4 diagnostics-10-00711-t004:** FBUPs results for ADDE and short FBUT-DE.

**FBUPs**	**ADDE**	**Short FBUT-DE**
	**Area, Line**	**Spot, Dimple**
Sensitivity	0.747	0.416
Specificity	0.541	0.794
Pre-test probability	0.466	0.299
Post-test probability	0.587	0.463
Positive likelihood ratio	1.627	2.018

ADDE = aqueous deficient dry eye; FBUPs = fluorescein breakup patterns; short FBUT-DE = short fluorescein breakup time dry eye.

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
