# Peer review of "Characteristics and Utility of Fluorescein Breakup Patterns among Dry Eyes in Clinic-Based Settings"

_diagnostics, 2020, doi:10.3390/diagnostics10090711_

Round 1

Reviewer 1 Report

the attached file 

Author Response

Reviewer 1

This is an interesting cross-sectional study regarding the characteristics and utility of fluorescein breakup patterns among patients with dry eyes in clinic-based settings. However, there are some issues must be addressed before it can be considered for publication.  

We thank the reviewer for the time and effort spent reviewing our paper. We have revised our manuscript, including the Results section, taking into account the reviewer’s insightful comments.

 Major comments:

  1. In lines 79-80, the total number of this study seems to be 500. However, it is not clear and consistent with the numbers shown in Figure 1. Please clarify it.

Response: I would like to show my apology for the complication. We have set the protocol of registering up to 50 patients at each clinic. However, some clinics did not reach registering 50 patients, and 463 patients were initially registered. A little is described in Figure 1, however, since I will be deleting Figure 1 due to the minor comment (2: duplication of information in Figure 1), I have inserted a description

Page 4 line 158

Forty-five to 50 patients were enrolled from each site and a total of 926 eyes of 463 patients were initially registered (Table 1).

  1. In lines 149-150, the authors described that the likelihood ratio was used to calculate posttest probabilities for ADDE and short FBUT-DE subgroups, please provide a more detailed procedure for this analysis. Moreover, where are the relevant results in which table(s) or figure(s)?

Response:

Thank you for your precious comment. Table 3 (changed from originally Figure 5) shows the relevant data. The 74.7% of ADDE were consisted from line break and area break, and the 42.4% of short FBUT-DE were consisted from spot break and dimple break. However, I would like to add a Table 4. To describe the calculation of likelihood ratio.

Page 7 line 240

Table 4. FBUPs results for ADDE and short FBUT-DE

FBUPs

ADDE

short FBUT-DE

area, line

spot, dimple

Sensitivity

0.747

0.416

Specificity

0.541

0.794

Pre-test probability

0.466

0.299

Post-test probability

0.587

0.463

Positive likelihood ratio

1.627

2.018

ADDE = aqueous-deficient dry eye; FBUPs = fluorescein breakup patterns; short FBUT-DE = short fluorescein breakup time dry eye

  1. In Figure 1, please explain why 28 eyes (14 patients) were excluded. In addition, the number in the last box indicated that FBUPs of 867 eyes (439 patients) were classified, please describe how to get the number of 439 ?

Response: I would like to show my apology for the complication. As written in line 161, 28 eyes of 14 patients were excluded since nine cases was disqualified (not dry eye), three patients were <20 years of age, and two patients withdrew consent. In addition, since the FBUPs of 31 eyes of 21 patients (20 eyes of 10 patients and 11 eyes of 11 patients) were not recorded, 867 eyes of 439 patients were enrolled in the final analysis. I have added a description.

Page 4 line 162

In addition, records of FBUPs were not recorded in 31 eyes of 21 patients (20 eyes of 10 patients and 11 eyes of 11 patients).

  1. In Table 1, most of study subjects are female (over 80%), the authors should discuss whether the gender effect can affect relevant analyses in this study.

Response:

I would like to show my appreciation for your insightful comment. Although the prevalence of dry eye disease is known to be higher in women compared to men, the rate of women was high and covered 86.4% in this study. To see the gender difference in the distribution of FBUPs, I have classified FBUPs to man and woman as below. The results were similar between man and woman, and the order of the FBUPs were the same; line break (man: 54.2%, woman: 51.5%), dimple break (man: 12.7%, woman: 16.9%), random break (man: 11.9%, woman: 13.8%), spot break (man: 6.8%, woman: 11.1%), area break (man: 5.9%, woman: 4.4%) and undefined (man: 8.5%, woman: 2.7%), respectively. I did not perform further assessment; however, it seems that the order of frequently diagnosed FBUPs seems to match between man and woman. 

All

Man

Woman

eyes

(%)

eyes

(%)

eyes

(%)

All

867

100

118

100

749

100

Line break

448

51.7

64

54.2

384

51.5

Area break

40

4.6

7

5.9

33

4.4

Spot break

91

10.5

8

6.8

83

11.1

Dimple break

141

16.3

15

12.7

126

16.9

Random break

117

13.5

14

11.9

103

13.8

Undefined

30

3.5

10

8.5

20

2.7

I have added a description in the “Discussion”.

Page 9 line 314

The rate of women was high and covered 86.4% in this study. Although the prevalence of dry eye disease is known to be higher in women compared to men, further analysis was performed to see the gender differences in FBUPs. The results were similar among the groups, and the order of the FBUPs were the same; line break (man: 54.2%, woman: 51.5%), dimple break (man: 12.7%, woman: 16.9%), random break (man: 11.9%, woman: 13.8%), spot break (man: 6.8%, woman: 11.1%), area break (man: 5.9%, woman: 4.4%) and undefined (man: 8.5%, woman: 2.7%), respectively. It resulted that the order of frequently diagnosed FBUPs seems to match between man and woman.   

  1. In lines 297-301, please indicate where are the relevant data?

Response:

I would like to show my apology for lack of information. Table 3 (changed from originally Figure 5) shows the relevant data. The 74.7% of ADDE were consisted from line break and area break, and the 42.4% of short FBUT-DE were consisted from spot break and dimple break. However, I provided a new Table 4 for more information.

Page 7 line 229

Pretest probabilities, posttest probabilities, and positive likelihood ratio were 46.6%, 58.7%, and 1.63, respectively, for ADDE (LB and AB); and 29.9%, 46.3%, and 2.02, respectively, for short FBUT-DE (SB and DB) (Table 4).

Minor issues:

  1. All the abbreviations such as DECS-J and DEWS must be present in the full form.

Response:

Thank you for your indication. I have spelled out the abbreviations. Also, I would like to show my apology for the mistake of the place of spelling out the abbreviation.

Page 1 line 44      

Dry Eye Cross-Sectional Study in Japan (DECS-J).

Due to the changes, “DECS-J” spelled out in line 68 was deleted in line.

Page 3 line 120    

The Schirmer I test was selected as it is regarded as a standardized test that provides an estimation of stimulated reflex tear flow, according to the Dry Eye WorkShop (DEWS) reports [14].

  1. The information in Figure 1 is a part of results shown in Table 1, therefore, the Figure 1 can be deleted.

Response:

Thank you for your comment. Due to the duplication, I have deleted Figure 1 and included the description in the “Study Population”. Accordingly, the number of the Figure has been changed.

  1. The descriptions regarding the Figure 3 are difficult to read. The authors can use

Fig. 3A, 3B, 3C and 3D to make the relevant contents clearer.  

Response:

Thank you for your kindness. I have used Fig. 2A, 2B, 2C and 2D in the description of  “3.2. FBUPs in Relation to Tear Function Tests and Ocular Surface Evaluation” section (page 6 line 182-202) (The numbering of Figure 3 was changed to Figure 2).

  1. Figures 4 and 5 are suggested to be replaced by Tables.

Response:

Thank you for your indication. I have changed Figure 4 and 5 to Table 2 and 3.

Table 2. Tear break up patterns and dry eye subgroups. Area break and line break were consisted from majority of Sjögren syndrome and aqueous deficiency, covered 75.0% and 57.2% respectively. Spot break and dimple break were consisted from majority of short FBUT-DE, covered 38.3% and 51.9%, respectively. Random break were consisted from majority of MGD, CL related and friction related DED, covered 50.7%.

Sjögren syndrome

Aqueous deficiency

short FBUT

MGD

CL

Friction

Line break

10.2

47.0

26.9

5.6

6.3

4.2

Area break

15.0

60.0

5.0

2.5

12.5

5.0

Spot break

8.6

30.9

38.3

6.2

7.4

8.6

Dimple break

1.5

17.8

51.9

13.3

8.9

6.7

Random break

6.0

25.3

18.1

16.9

18.1

15.7

Undefined

8.0

32.0

16.0

16.0

8.0

20.0

DED = dry eye disease; CL = contact lens; MGD = meibomian gland dysfunction; short FBUT-DE = short fluorescein breakup time dry eye. The results are shown in percentage (%).

Table 3. Tear break up patterns of aqueous deficient dry eye and short fluorescein breakup time dry eye subgroups. The 74.7% of ADDE were consisted from line break and area break. The 42.4% of Short FBUT-DE were consisted from spot break and dimple break.

Area break

Line break

Spot break

Dimple break

Random break

Undefined

ADDE

8.1

66.6

8.6

7.0

7.0

2.7

short FBUT-DE

0.8

48.7

13.0

29.4

6.3

1.7

All

5.0

54.3

10.2

17.0

10.4

3.1

ADDE = aqueous deficient dry eye; short FBUT-DE = short fluorescein breakup time dry eye. The results are shown in percentage (%).

Reviewer 2 Report

The assessment of fluorescein break up patterns is a very interesting approach to analyse further dry eye disease.

However, the methodology of this study has to be questioned:

Line 55: The authors mention that non-invasive TBUT (NIBUT) has been recommended for the measurement of TBUT, and arguably NIBUT represents the gold-standard for tear film break up. It is therefore surprising that they have not considered to include this non-invasive measurement as well as the lipid layer assessment in the measurement protocol for this study, which would have enabled a more accurate confirmation of the distribution of FBUP into different dry eye disease subgroups. As this represents an important limitation to this study, it should be mentioned in the discussion part.

2.2.1 – tear function tests and ocular surface evaluation:

- Was the amount of saline used to wet the test strips standardized? Did all test sites use the same brand of test strips? The fluorescein concentration varies between different manufacturers.

 - Three FBUT measurements were carried out on each eye – what was the time interval between each measurement? The measurement itself destabilises the tear film, so if the time interval is not kept long enough, the resulting FBUT will be shorter. How often was fluorescein applied? For each measurement? Was the order of eyes randomized?

- Were FBUPs assessed after or during the FBUT measurements? If they were assessed after the three FBUT measurements on each eye, the tear film would / could have been considerably destabilized – is this then still a realistic measurement?

- Caution has to be taken when both eyes of the subjects are considered for statistical analysis. What would be the benefit? Shouldn’t the results be similar for the two eyes? Indeed, with three FBUT measurements per eye, the tear film may be considerably destabilized in the second eye. Can the authors include the separate results for the two eyes? Were they different?

The fluorescein breakup patterns in figure 2 are very small – can the authors include larger photographs of the different patterns?

Author Response

Reviewer 2

We would like to express our appreciation to the reviewer for these constructive comments, which have addressed in the revised manuscript.

Comments and Suggestions for Authors

The assessment of fluorescein break up patterns is a very interesting approach to analyse further dry eye disease. However, the methodology of this study has to be questioned:

Line 55: The authors mention that non-invasive TBUT (NIBUT) has been recommended for the measurement of TBUT, and arguably NIBUT represents the gold-standard for tear film break up. It is therefore surprising that they have not considered to include this non-invasive measurement as well as the lipid layer assessment in the measurement protocol for this study, which would have enabled a more accurate confirmation of the distribution of FBUP into different dry eye disease subgroups. As this represents an important limitation to this study, it should be mentioned in the discussion part.

Response:

Thank you for your insightful comment. As written in DEWSⅡreport, we do agree that the tear film stability may affect the tear stability and NIBUT may have more less effects. However, it is also written in the DEWSⅡreport, that not all the clinics have the device to measure NIBUT, and FBUT still remains as one of the most commonly used diagnostic test for clinical practice. Since not all the clinics had the device to measure NIBUT, we were not able to measure NIBUT in this study. I have added a description in the “Discussion”.

Page 9 line 321

Secondly, although NIBUT is recommended to measure TBUT, we were not able to measure them since not all the clinics had the device.

2.2.1 – tear function tests and ocular surface evaluation:

- Was the amount of saline used to wet the test strips standardized? Did all test sites use the same brand of test strips? The fluorescein concentration varies between different manufacturers.

Response:

Thank you very for your comment. Standardization of the volume instilled is important, and two drops of saline solution was used on the fluorescein strip and shake off the excessive saline, not to increase the tear volume. We followed the procedure for fluorescein staining recommended by the Japanese Dry Eye Society. In Japan, a 50 mm2 fluorescein-impregnated strip is widely used in clinical practice and we have used the same brand (AYUMI Pharmaceutical Corporation, Tokyo, Japan), as written in the method. I have added a description about the amount of saline in the “Method”.

Page 3 line 93

Test strips containing fluorescein sodium (Fluores Ocular Examination Test Paper, Ayumi Pharmaceutical Co., Tokyo, Japan) were used for vital staining and FBUT measurement. Two drops of saline solution was used on the fluorescein strip and after shaking off the excessive saline, the tip of a wetted test strip had been applied once to the inferior temporal tear meniscus, and patients were requested to blink several times to ensure adequate mixing of the fluorescein dye with tears.

 - Three FBUT measurements were carried out on each eye – what was the time interval between each measurement? The measurement itself destabilises the tear film, so if the time interval is not kept long enough, the resulting FBUT will be shorter. How often was fluorescein applied? For each measurement? Was the order of eyes randomized?

Response:

I would like to show my appreciation for your insightful comment. As indicated, to diagnose FBUPs effectively should be taken in care. Yokoi et al. reported the procedure in detail in AJO (2017) and IOVS (2018). The essential point is not to increase tear volume and instruct the patient to briskly open the eyes gently. Reproducible FBUPs are regarded as important, which is based on ocular surface pathophysiology. FBUPs were diagnosed during the usual procedure of measuring FBUT. As indicated in the DEWS Ⅱ report (2017), after instillation of fluorescein, FBUT is instructed to be measured after 3 natural blinks, and we followed this procedure. Fluorescein was applied only once before the first measurement, and the patients were instructed to blink 3 times in each interval. I would like to show my apology not to be able to state the accurate time, however, by following this procedure, among the 867 eyes included in the study, FBUPs of 837 eyes (96.5%) had reproducibility, and were able to classify. The measurement was performed from the right eye to the left eye. I have added a description in the “Method”.

Page 3 line 94

Two drops of saline solution was used on the fluorescein strip and after shaking off the excessive saline, the tip of a wetted test strip had been applied once to the inferior temporal tear meniscus, and patients were requested to blink several times to ensure adequate mixing of the fluorescein dye with tears. The time interval between the last complete blink and the appearance of the first corneal dark spot was measured with a stopwatch. Three natural blinks were performed between the each measurement, and the mean of three measurements was recorded as FBUT, assessing FBUPs at the same time. The measurement was performed from the right eye to the left eye.

- Were FBUPs assessed after or during the FBUT measurements? If they were assessed after the three FBUT measurements on each eye, the tear film would / could have been considerably destabilized – is this then still a realistic measurement?

Response:

Thank you for your indication. FBUPs were assessed during the FBUT, since FBUPs are seen during the fluorescein tear breakup. Recurrence of breakup in the same corneal region is reported, indicating that the position of breakup was nonrandom but was related to corneal properties and heterogeneity (Yokoi, 2017; Liu, 2006). We think classification of FBUPs is a simple method which can be performed in clinics, and which is based on ocular surface pathophysiology. I have added a description in “Method”.

Page 3 line 99

Three natural blinks were performed between the each measurement, and the mean of three measurements was recorded as FBUT, assessing FBUPs at the same time. The measurement was performed from the right eye to the left eye.

- Caution has to be taken when both eyes of the subjects are considered for statistical analysis. What would be the benefit? Shouldn’t the results be similar for the two eyes? Indeed, with three FBUT measurements per eye, the tear film may be considerably destabilized in the second eye. Can the authors include the separate results for the two eyes? Were they different?

Response:

I would like to show my appreciation for the precious comment. Generalized linear mixed model was performed since two eyes (right and the left eye) of one patient has a single age (interval scale) and a single gender (nominal scale). Only the statistical analysis including age and gender resulted to take in consideration of both eyes (Table 1). Below is the result of the t-test analysis counting two eyes per patient. The statistically significant results were the same as the result of generalized linear mixed model in this analysis.

Age (yrs)

Gender

P value (vs Line)

P value (vs Line)

Area break

0.69

0.58

Spot break

<0.0001

0.16

Dimple break

0.07

0.27

Random break

<0.001

0.52

Undefined

0.99

<0.01

The fluorescein breakup patterns in figure 2 are very small – can the authors include larger photographs of the different patterns?

Response:

Thank you for your indication. I have changed the original Figure 2 with larger photographs.

Page 5 line 175

Round 2

Reviewer 1 Report

1 The table is confusing to the readers

  consider present  the data of subgroup only.

2 The title of the table 2 and table 3 should be revised. 

   The description of the data should be deleted, and put the description in the result section of text. 

Author Response

Reviewer 1

We would like to express our appreciation to the reviewer for these constructive comments.

1 The table is confusing to the readers

consider present the data of subgroup only.

Response:

I would like to show my full apology for the complexity of the tables. I have looked the data of all the tables and the data of the “All” in the table 3 seems better to be deleted and are made simplified.

Page 7 line 230

Table 3. Tear breakup patterns of aqueous deficient dry eye and short fluorescein breakup time dry eye subgroups.

Area break

Line break

Spot break

Dimple break

Random break

Undefined

ADDE

8.1

66.6

8.6

7.0

7.0

2.7

short FBUT-DE

0.8

48.7

13.0

29.4

6.3

1.7

ADDE = aqueous deficient dry eye; short FBUT-DE = short fluorescein breakup time dry eye. The results are shown in percentage (%).

2 The title of the table 2 and table 3 should be revised. 

   The description of the data should be deleted, and put the description in the result section of text. 

Response:

I would like to show my appreciation for the precious advice. Since the same description is written in the result section, I have deleted the description of the data in the table 2 and table 3.

Reviewer 2 Report

Although it is understandable for practical / logistic reasons that tear break up time was not measured non-invasively, I would have appreciated this being mentioned specifically as a limitation of this study in the discussion session.

Author Response

Reviewer 2

Although it is understandable for practical / logistic reasons that tear break up time was not measured non-invasively, I would have appreciated this being mentioned specifically as a limitation of this study in the discussion session.

Response:

We would like to thank the reviewer for the time and effort spent reviewing our paper.

I would like to show my appreciation for your precious comment. I have mentioned the use of FBUT as a limitation specifically, in the discussion session.

Page 9 line 315

Secondly, although automated NIBUT is recommended to measure TBUT rather than subjective FBUT by the observer [16], we were not able to measure them since not all the clinics had the device. Much attention was taken not to influence the tear film stability during the FBUT measurements; however, we must take in mind that NIBUT is known to be less affective.